# FIRST-EXPLORE, THEN EXPLOIT: META-LEARNING INTELLIGENT EXPLORATION

## ABSTRACT

Standard reinforcement learning (RL) agents never intelligently explore like a human (i.e. by taking into account complex domain priors and previous explorations). Even the most basic intelligent exploration strategies such as exhaustive search are only inefficiently or poorly approximated by approaches such as novelty search or intrinsic motivation, let alone more complicated strategies like learning new skills, climbing stairs, opening doors, or conducting experiments. This lack of intelligent exploration limits sample efficiency and prevents solving hard exploration domains. We argue a core barrier prohibiting many RL approaches from learning intelligent exploration is that the methods attempt to explore and exploit simultaneously, which harms both exploration and exploitation as the goals often conflict. We propose a novel meta-RL framework (First-Explore) with two policies: one policy learns to *only* explore and one policy learns to *only* exploit. Once trained, we can then explore with the explore policy, for as long as desired, and then exploit based on all the information gained during exploration. This approach avoids the conflict of trying to do both exploration and exploitation at once. We demonstrate that First-Explore can learn intelligent exploration strategies such as exhaustive search and more, and that it outperforms dominant standard RL and meta-RL approaches on domains where exploration requires sacrificing reward. Surprisingly and importantly, on such domains, First-Explore not only achieves higher final episode reward, it also achieves higher *cumulative* reward. First-Explore is a significant step towards creating meta-RL algorithms capable of learning human-level exploration, which is essential to solve challenging unseen hard-exploration domains.

## 1 INTRODUCTION

Reinforcement learning (RL) performs challenging tasks, including plasma control (Degrave et al., 2022), molecule design (Zhou et al., 2019a), game playing (Silver et al., 2018), and controlling robots (OpenAI et al., 2019). Despite this success, standard RL is *very* sample inefficient. Agents take hundreds of thousands of episodes of play to learn a task that humans could learn in a few tries (Mnih et al., 2013).

This sample inefficiency has several causes. First, standard RL cannot condition on a complex prior such as a human's common sense or general experience (Pertsch et al., 2021) (e.g., knowing before training on a game that e.g., one can collect coins to obtain reward). It has been shown that much of the sample efficiency of humans comes from such priors (Dubey et al., 2018). Second, a standard RL agent cannot remember previous episodes, and only learns via slow weight updates. Having no memory prohibits high sample efficiency. Memory is required to e.g., navigate to a treasure location after reading a treasure map last episode, or e.g., immediately copy an opponent's chess opening after losing to it. Third, standard RL and standard meta reinforcement learning (meta-RL) both use the same policy to explore (gather data to improve the policy) and to exploit (achieve high episode reward) (Schulman et al., 2017; Adaptive Agent Team et al., 2023). Using the same policy for both purposes can lead to terrible performance when good exploration and good exploitation require very different behaviors (e.g., when exploration requires sacrificing episode reward). Sec. 5.2 shows standard meta-RL can fail to train when deep sacrificial exploration is required.

We present *First-Explore*, a simple framework for meta-RL that overcomes these limitations by learning a pair of policies: an explore policy that can *intelligently* explore, and an exploit policy that can *intelligently* exploit. First-Explore enables the potential of learning policies that exhibit meta-RL's human-level-in-context-sample-efficient learning on unseen hard-exploration domains including hostile ones that require sacrificing reward to properly explore.

## 2   RL ISSUES

**Exploring by Exploiting:** Standard RL uses a single policy for two different purposes: i) Exploring: gathering data to improve the policy and ii) Exploiting: achieving high episode reward (Sutton et al., 2018). Standard RL algorithms (such as PPO (Schulman et al., 2017)) rely on exploring by sampling the small area of policy space covered by a noisy policy centered on exploitation, e.g., by ensuring the exploit policy has high entropy (Haarnoja et al., 2018) or by epsilon-greedy sampling of the policy (Mnih et al., 2013).

Exploring by relying on such *noisy exploiting* struggles in the presence of a reward signal that does not always lead to the global optima, i.e., deceptive reward (e.g., if the environment is very harsh an agent may learn to stay still so as to avoid the penalty of moving incorrectly). In the presence of deceptive reward good exploration must *sacrifice* episode reward. We categorize exploration by whether it makes such sacrifice:

* *Sacrificial* **exploration**: exploration that is *not* exploitative is *sacrificial* as one is 'sacrificing' episode reward for information gain. **Examples**: paying for information or tutoring, doing practice drills, practicing ones weaknesses, attempting a *new* strategy while a previously tried one works.

* *Coincidental* **exploration**: exploitation that happens by coincidence when *noisily exploiting*(exploiting with noise potentially added or encouraged). Relying on *coincidental* exploration is the standard RL approach, and is vulnerable to local optima. **Examples**: practicing one's strengths, playing normal matches, attempting a *new* strategy when all previously tried ones fail.

Standard RL never intentionally *sacrificially* explores because each episode is spent trying to maximize reward. This inability prevents standard RL from optimally exploring, and so causes greater sample inefficiency, making solving hostile tasks (where exploration requires sacrificing reward) infeasible.

**Memory-less Exploration:** A standard RL agent has no knowledge or memory of previous episodes, and so (while noisily exploiting) it will do approximately the same 'exploration' repeatedly. This repetition can make standard RL exploration hugely sample inefficient. While the agent's policy may change due to updates to the policy's weights, the policy change is slow, and unlikely to allow human-level adaption, wherein people change their policy substantially and appropriately based on a single episode of experience (e.g., exploring new territory in each episode).

**No Prior on Exploration:** Effective and efficient exploration *requires* a prior on how to explore in the environment (e.g., intuiting that levers might open doors) (Dubey et al., 2018). Further, a good exploration prior is often different from a good exploitation prior because optimal exploration often requires sacrificing episode reward, e.g., to experiment with new strategies.

Imagine playing an adventure game: each episode one explores a fixed dungeon full of treasures and traps. In the dungeon one finds a new (untested) lever that likely triggers a trap (huge penalty), but may unlock a new room with great treasure (reward). If one is purely exploiting (maximizing current episode reward, e.g., for a high score) then one should not risk the lever as the trap chance out weighs the potential treasure gain. However, while exploring it is future episode reward that matters (not current), and the slim chance of treasure is worth it (as with the fixed dungeon it can be obtained every subsequent episode). Further, if the lever is a trap then it be will triggered once and then avoided. The only concern, while exploring, is triggered trap opportunity cost potentially prohibiting further exploration this episode. Both ways of playing corresponds to a prior on how the player should act (use unfamiliar levers or don't), however the prior for exploitation actively prohibits effective exploration (levers remain untried).

## 3 META-RL AND RELATED WORK

Meta-RL addresses standard-RL issues by learning a reinforcement learning algorithm: a map from a context of rollouts $c$ in an environment $m$ to a peformant policy $\pi_{\theta,c}$ specialized to that environment. Any architecture capable of memory can be used (e.g., transformers (Adaptive Agent Team et al., 2023) or recurrent neural networks (Wang et al., 2016)). To train meta-RL, one specifies a distribution of environments $\mathcal{M}$. Each training step, environments are sampled, and the agent plays a sequence of same-environment episodes. Because the agent has memory of earlier current-sequence episodes, it can learn to adapt to the current environment $m$. It can also learn the prior that the environment comes from the training distribution, $m \sim \mathcal{M}$.

*Once trained*, the learnt RL algorithm can be very sample efficient (at in-context adaptation) (Adaptive Agent Team et al., 2023; Duan et al., 2016). For example, when trained to find a reward location in mazes, a learnt RL algorithm can avoid maze areas already ruled out by past exploration (Duan et al., 2016). This capability allows an unseen maze to be solved in a few tries, which is fewer episode rollouts than are needed for a typical standard RL gradient update (Duan et al., 2016).

**Cummulative Reward Meta-RL:** Methods such as $RL^2$ (Duan et al., 2016; Wang et al., 2016), VariBAD (Zintgraf et al., 2019), HyperX (Zintgraf et al., 2021), AdA (Adaptive Agent Team et al., 2023) train a single policy to maximize (possibly discounted) cumulative reward. Maximizing cumulative reward allows some sacrificial exploration (trading off low reward in initial episodes for higher reward in later ones). However, because each episode reward matters to the final sum, sufficiently deceptive rewards can still prevent good exploration. Sec. 5.2 shows this liability can result in these methods having significantly lower cumulative reward than First-Explore.

**Meta-RL Exploration:** Methods such as MetaCURE (Zhang et al., 2021), EPI (Zhou et al., 2019b) and CCM (Fu et al., 2021) learn an exploration policy that aims to extract maximum environment information (independent of whether such information informs good exploitation). While decoupling exploration from exploitation guards against deceptive reward preventing exploration, these approaches discard grounding exploration in (maximizing) future reward. Not grounding exploration in future exploitation reward means that time may be spent learning irrelevant information (e.g., the exact penalty of bad actions). This distraction potentially prevents good exploration.

E-$RL^2$ (Stadie et al., 2018) modifies $RL^2$ to ignore the first-$k$ episode rewards. This modification enables pure sacrificial exploration. However, E-$RL^2$ introduces an across-episode value assignment problem: identifying which exploration episodes enabled good subsequent exploitation. This problem potentially limits training sample efficiency. Further, the exploratory episodes number $k$ is set as a hyperparameter and constant across all tasks (both at training and at inference), preventing efficient combination with a curriculum that contains different difficulty tasks (as hard tasks may need significantly more exploration episodes than easy ones). Finally, hard coding $k$ limits the flexibility and usefulness of E-$RL^2$ because one cannot explore until a satisfactory policy quality is reached, preventing meta-RL in-context adaptation from off-the-shelf replacing standard RL. While, like E-$RL^2$, not discouraging sacrificial exploration, First-Explore addresses these limitations.

DREAM (Liu et al., 2021) also separately optimizes exploration and exploitation policies (and grounds exploration in exploitation), but has four complex, manually designed, interacting components and a reliance on knowing unique problem IDs during meta-training. This complexity enables increased sample-efficiency by avoiding the chicken and egg problem of simultaneously learning explore and exploit policies. First-Explore shows that such complexity is unnecessary, at least for the domains we tested in. Unlike E-$RL^2$, because a part of DREAM's machinery must learn to produce the right information per problem based on the (unique, random) problem ID only, it is unable to generalize or handle never-seen-before challenges during meta-training, raising questions about its scalability and generality. For example, DREAM may potentially be difficult to apply to problems where each training environment is unique (e.g., for environments with continuous variables, or samples from otherwise vast search spaces). It may also struggle when each environment is a hard-exploration challenge, as it may be difficult for the model to explore enough to learn which information is required to solve the problem. We believe curricula are necessary to solve such environments. However, because DREAM cannot generalize during meta-training (as described above), it cannot take advantage of a curriculum to build an exploration skill set to tackle harder and harder exploration challenges, unlike First-Explore.

### 3.1 OTHER WORKS ADDRESSING EXPLORATION

A rich non-meta-RL exploration literature exists. E.g., Intrinsic Motivation (IM) replaces the environment reward with an intrinsic motivation reward such as novelty (Aubret et al., 2019), and so enables *sacrificial* exploration. However, non-meta-RL methods are limited by being slow to adapt due to lacking a memory not encoded via weights (sec. 2) and not having a complex learnt prior on exploration (sec. 2). Further, many of these methods enable *sacrificial* exploration by entirely ignoring the reward signal, leading to pathologies such as the noisy TV problem (Burda et al., 2018; Ladosz et al., 2022), where an agent looking for new states will find a TV showing white noise to be endlessly captivating. Another method, Go-Explore (Ecoffet et al., 2019), decouples exploration and exploitation, but lacks complex priors.

There are also MPC methods (e.g., Mehta et al. (2022)) and approaches within the multi-armed bandit literature, and regret-based learning (Ladosz et al., 2022). However, all the methods have issues (e.g., computationally infeasible on complex long-horizon environments, requiring human-coded priors, etc... Only meta-RL is both a) computationally tractable and b) can potentially achieve human-level sample efficiency in complex environments. The First-Explore framework builds on this promise by allowing meta-RL to readily sacrificially explore.

## 4 FIRST-EXPLORE FRAMEWORK

The First-Explore Framework overcomes the discussed limitations by learning a pair of policies. An explore policy $\pi_{\text{explore},\theta,c}$ that explores and provides information (context) for itself and for exploitation, and an exploit policy $\pi_{\text{exploit},\theta,c}$ that exploits after every explore providing feedback to train the explore policy. The policies may share or have separate parameters, e.g., for policies with separate parameters, one could write $\theta = (\theta_{\text{explore}}, \theta_{\text{exploit}})$ with each policy only dependent on its own subset of $\theta$. This framework is visualized in Figure 1.

- The explore policy $\pi_{\text{explore},\theta}|c$ gathers informative environment rollouts based on the current context $c$ (all previous explores) and parameters $\theta$.

- The exploit policy $\pi_{\text{exploit},\theta}|c$ exploits (maximizes episode return) based on the current context $c$ (all previous explores) and parameters $\theta$.

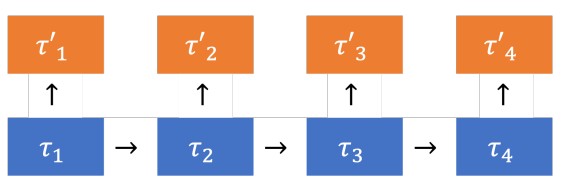

Figure 1: The **First-Explore Framework**. Arrows depict the flow of context / memory. The explore episodes $\tau_1, \ldots, \tau_4$ in blue are purely exploratory and are able to *sacrificially* explore. They are sampled using the previous-explore-conditioned explore policy $\tau_t \sim \pi_{\text{explore},\theta}|\{\tau_1, \ldots \tau_{t-1}\}$. The exploit episodes $\tau'_1, \ldots, \tau'_4$ in orange are purely exploitative, and are sampled using the previous-explore conditioned exploit policy $\tau'_t \sim \pi_{\text{exploit},\theta}|\{\tau_1, \ldots \tau_t\}$.

On complex tasks, the policies need to be learnt together. Learning together is necessary as each policy is limited by the other's quality. Provided poor context, it may be hard to distinguish excellent from poor exploitation (e.g., with no useful information good exploitation may be impossible). Similarly, if the exploitation policy ignores context then all exploration has the same value (no value).

A central idea of First-Explore is that the exploratory value $v_{\text{explore}}$ of an explore episode $\tau_t$ given a context of past episodes $\{\tau_1, \ldots, \tau_{t-1}\}$ is the increase in expected reward of a subsequent exploit when the explore episode is added to the context to create new context $\{\tau_1, \ldots, \tau_{t-1}, \tau_t\}$. $v_{\text{explore}}(\tau_t)|\{\tau_1, \ldots, \tau_{t-1}\} = \mathbb{E}(\tau_{\text{exploit}}|\{\tau_1, \ldots, \tau_{t-1}, \tau_t\}) - \mathbb{E}(\tau_{\text{exploit}}|\{\tau_1, \ldots, \tau_{t-1}\})$ where $\tau_{\text{exploit}}|c \sim \pi_{\text{exploit},\theta}|c$. As only the $\mathbb{E}(\tau_{\text{exploit}}|\{\tau_1, \ldots, \tau_{t-1}, \tau_t\})$ term depends on the explore episode $\tau_t$, the training reward for the explore policy can be simplified to be just the reward of the following exploit episode $\tau_{\text{exploit}}|\{\tau_1, \ldots, \tau_{t-1}, \tau_t\}$.

First-Explore trains by performing rollouts as depicted in Figure 1, with the explore policy being optimized to maximize the reward of the subsequent exploit, and the exploit policy being optimized

to maximize each exploit episode reward. First-Explore can be combined with different meta-RL approaches and losses.

Once trained, First-Explore (in-context) adapts to an environment by performing iterated exploration rollouts using the exploration policy, building the context. As the agent explores, the quality of exploitation improves as the agent has more environment information. Exploration rollouts become the analogue of standard-RL training, and exploitation rollouts the analogue of standard-RL evaluation. As in standard-RL, one might explore (train) until a the environment is solve (a desired exploit quality is reached), or explore (train) for a set number of rollouts (i.e., train epochs).

## 5 RESULTS

First-Explore results used a GPT-2 style transformer architecture (Radford et al., 2019). For simplicity, the parameters are shared between the two policies, differing only by a final linear-layer head. We use a novel loss based on predicting the sequence of future actions conditional on the episode having high reward, which preliminary experiments showed improved training stability. While an innovation, it is not core to the framework, and standard training (e.g., PPO) should work as replacement. The controls (VariBAD, $RL^2$, HyperX) were run using the VariBAD and HyperX official codebase. The controls are run for less time than First-Explore due to the controls converging and not exhibiting improvement with further training (unlike First-Explore which continued to improve). Full architecture, training details and hyperparameters are given in the appendix, along with plots demonstrating that the controls converge and a more detailed explanation of why they do.

All policies were trained ten times with ten different random seeds. Furthermore, the in-context learning of each training run was evaluated ten times each on an independently sampled batch of environments (for a total of 100 evaluations). Each treatment is then visualized by a line showing the mean over the evaluations and training runs.

A single evaluation of the trained policies involves sampling a large batch of environments, performing iterated rollouts in *each* environment (allowing the policy to in-context adapt to each environment) and calculating the average statistics across the batch (e.g., the average first episode exploit reward). The lighter-shaded area shows the minimum and maximum value (across evaluations and seeds). If the light area shaded around one line (e.g., the First-Explore exploit reward) is above the light shaded area around another (e.g., $RL^2$ reward) then, in all 100 evaluations, one treatment beats the other, which (as the runs are independent) is statistically significant ($p \leq 2^{-10}$). All lines have these shaded areas, however the deviation between evaluations can be so small that the shaded areas can be hard to see.

### 5.1 GAUSSIAN 10-ARMED BANDIT DOMAIN

A 10-armed Gaussian Bandit environment is specified by 10 arm means $\mu_{\{1,...10\}} \in \mathbb{R}$. Each time step $t$, the agent chooses an arm $a_t$ and receives reward $r_t$ equal to the arm mean plus a normally distributed noise term, $r_t = N(\mu_{a_t}, \frac{1}{2})$. A meta-RL agent also observes its previous actions and their rewards, and can adapt based on that. Each environment's arm means are normally distributed, $\mu_{\{1,...,k\}} \sim N(\mathbf{0}, \mathbf{I})$. After training, we compare First-Explore to the following: i) UCB1. UCB1 estimates an upper confidence bound and selects the arm that maximizes $\text{ucb}_i(t) = \hat{\mu}_i(t-1) + c\sqrt{\frac{2\log t}{T_i(t-1)}}$ where $\hat{\mu}_i(t-1)$ is the estimated mean reward of the ith arm, $T_i(t-1)$ is the number of times the ith arm has been pulled, and $c$ is a tunable hyperparameter. Similarly to Duan et al. (2016), we initialize UCB1 with a prior, corresponding to one observation of zero for each arm. ii) Thompson Sampling (TS) (Thompson, 1933), which samples arm means from the posterior distribution, and chooses the arm with the best sampled mean. iii) $RL^2$: See appendix for training details. iv) Round Robin Arm Selection, which samples the arms in fixed order from 1 to 10 repeating. v) Random Arm Selection. We compare three policy properties: a) how well the policies cover the arms (sample each arm at least once) (Fig. 2 A b) the average reward of a policy (Fig. 2 B1-2 at different scales) and (c) how well the exploration of a policy informs a hand-coded strategy of picking the arm with the highest mean reward seen (or an unseen arm if all seen arms have negative sample means) (Fig. 2 C1-3 at different scales).

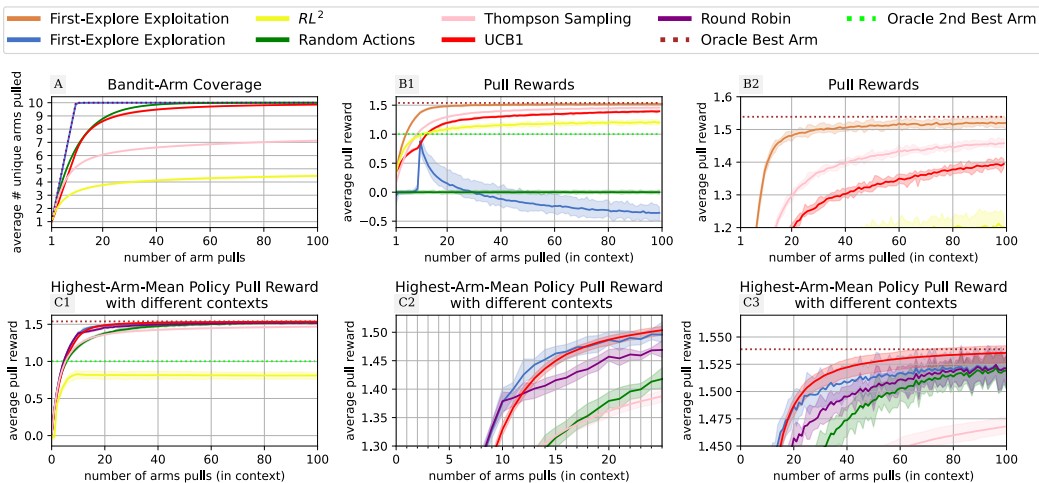

Figure 2: **A) Bandit-Coverage:** First-Explore Exploration (blue line) and Round Robin (dotted purple) are the only policies to consistently try all arms in the first ten tries (and this best informs exploitation see C2. Random selection (green) takes significantly longer (but is $\approx 10$ eventually), UBC1 (red) is worse than random (9.75 average coverage after 100 pulls) and TS and $RL^2$ are the worst ($< 8$ coverage). **B1-2) Differently Zoomed Pull Rewards:** First-Explore Exploitation (orange) outperforms all other policies (quickly nearing oracle performance (dotted brown)). **C1-3) Differently Zoomed Hand-coded Exploit Reward with Policy Exploration:** First-Explore Exploration and Round Robin (C2 purple overlapping blue) provide the best exploration in the first ten pulls, after which First-Explore Exploration outperforms Round Robin (C2 blue above purple) (by focusing on high expected value arms (see B1)) to be eventually slightly outperformed by UBC1 (C3 red above blue). All policies other than $RL^2$ (yellow) and TS (pink) eventually inform Exploitation sufficiently to get very close to oracle performance (dotted brown). This insufficiency corresponds to $RL^2$ and TS having poor arm coverage (see A).

First-Explore learns intelligent exploration on this domain, learning a policy that exhaustively searches (Fig. 2 A blue line) in the first ten actions (almost always trying all ten arms in the first ten pulls). Once all arms have been tried once, First-Explore changes its exploration to sampling high reward arms (Fig. 2 B2 left blue line). This series of average pull rewards show how the learnt policy is grounded in reward (by focusing on high reward arms at times), while also able to *sacrificially* explore (by getting low expected reward for the first ten pulls).

Evaluating the quality of exploration by using their context to inform a hand-coded policy that picks the arm with highest seen mean (or an unseen arm if all seen arms are negative), First-Explore exploration provides better exploration than hand-coded round robin selection (Fig. 2 C2 blue line higher than purple line after ten pulls), while in the first 12 pulls outperforming UCB1 (Fig. 2 C2 red line) due to First-Explore learning to initially perform exhaustive search. UCB1 achieves slightly higher reward (1.537 vs 1.525) after many pulls (Fig. 2 C3), due to the tiny marginal gain of better informing the choice between arms with very similar (high) rewards.

First-Explore Exploitation significantly outperforms all other policies (Fig. 2 B1 orange line above all others), due to First-Explore decoupling exploration and exploitation, and after multiple pulls is close to oracle performance (Fig. 2 B1 brown line). While in this domain, it may be easy to program good exploitation (e.g., pick the arm with highest seen arm mean), this property is absent from more complex domains (e.g., driving a car given past episodes of training), and highlights the value of decoupling exploration and exploitation when problems naturally having low stakes training (e.g., flight simulation) and high stakes performance (e.g., flying an actual plane).

Interestingly, after the First-Explore exploration policy changes to sampling high reward arms, the explore pull rewards trend steadily downward and eventually become negative. This behavior may be because First-Explore learns to heavily weight reducing arm-uncertainty and (as it starts with the high-reward arms) eventually only lower reward arms have significant uncertainty left.

## 5.2 $9 \times 9$ DARK TREASURE-ROOM DOMAIN

$9 \times 9$ Dark Treasure-Rooms (inspired by the Darkroom in Laskin et al. (2022)) are $9 \times 9$ grids full of treasures and traps. The agent navigates (up, left, down, or right) to find treasure, and cannot see its surroundings. Only its current $(x, y)$ coordinates are observed, with a meta-RL agent also observing past coordinates, rewards and actions. Each environment has 8 objects (treasures or traps), and when the agent encounters a treasure or trap it consumes/activates it, and receives an associated reward (positive or negative). The treasure and trap values are uniformly distributed in the range $-4$ to 2 (i.e., $r_i \sim U[-4, 2]$). The locations of the objects are randomly sampled uniformly, with overlapping objects having their rewards/penalties summed. Importantly, the average value of any location is negative, meaning that visiting a new location gives a negative expected reward. Further, the exploratory value of a policy is well approximated by the number of unique coordinates visited, as each coordinate has a chance of containing treasure (and there is nothing to gain by visiting the same coordinate twice).

Because the agent lacks sight, to reliably find treasure while avoiding traps, a standard-RL agent must store the environment trap and reward locations in the agent's weights. This requirement makes each individual environment a standard-RL training challenge requiring many (thousands or more training episodes). In contrast, a meta-RL agent has access to a context of all past environment interactions, and so can instead with each episode in-context adapt to newly sampled environments, rather than needing to train anew. However, the negative expected reward for visiting new states makes the environment distribution hostile to *coincidental* exploration (exploring by noisy exploiting), thus requiring *sacrificial* exploration. To provide an environmental control that does not need *sacrificial* exploration, we also introduce a 'Kind-Room' domain with no traps ($r_i \sim U[0, 2]$) otherwise identical to the Dark Treasure-Room. First-Explore is compared to random actions selection and meta-RL algorithms $RL^2$, HyperX, and VariBAD (see related work section).

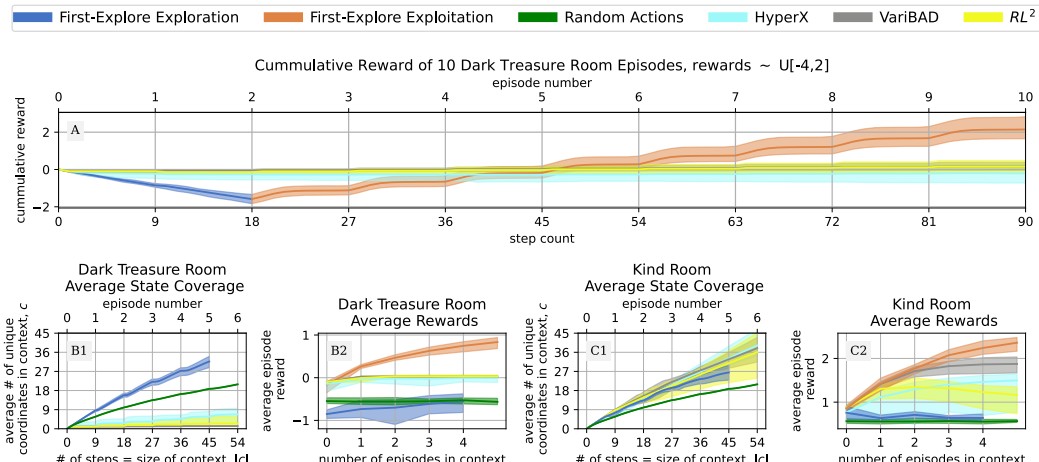

Figure 3: **A) 10 Episode Cumulative Reward:** First-Explore Exploration (blue) for two episodes, followed by First-Explore Exploitation (orange) conditioned on the frozen context of the two explorations, achieves significant positive average cumulative reward (on newly sampled Dark Treasure Rooms). HyperX (teal), VariBAD (grey), and $RL^2$ (yellow) fail to achieve significant positive reward, due to the hostility of
the space. **B1) Dark-Treasure Room Exploration:** VariBAD (grey) and $RL^2$ (yellow) learn to stay still, due to the hostility of the space, while HyperX (teal) (despite its exploration incentive) learns to avoid exploration covering less space than random action selection (green). **B2) Dark-Treasure Room Exploitation:** First-Explore Exploitation (orange) achieves significant positive reward, while the meta-RL controls (having learnt to stay still) achieve close to zero reward. The significant negative reward of the random play (green) and First-Explore Exploration highlights the hostility of the domain. **C1-2) Kind-Room Results:** When the environments only have positive rewards *coincidental* exploration is possible and all trained policies explore and exploit well. Despite this domain property, First-Explore exploitation C2 (orange) still outperforms the other methods, highlighting the value of decoupling exploration and exploitation.

The first experiment (Fig. 3 A) trains all methods on Dark-Treasure Rooms for a sequence of ten episodes each of nine steps. The meta-RL controls all fail to achieve significant cumulative reward despite a) training directly to do so, and b) significant cumulative reward being possible (Fig. 3 A teal, grey and yellow lines close to zero, despite orange line $\approx 2$ at step 90). Although not being trained for it, First-Explore achieves significant cumulative reward, with two explorations followed by repeated exploitation conditioned on the two explorations achieving significant average cumulative reward on newly sampled Dark-Treasure Rooms. The meta-RL controls learn to stay still (see below) preventing the methods from learning due to insufficient environment exploration while training (and so poor meta-exploration (see. for meta-exploration Beck et al. (2023)). This dynamic demonstrates that First-Explore can have a powerful meta-training advantage over other methods.

The second experiment (Fig. 3 B1-2 & C1-2) is sequences of six episodes each of nine steps. First-Explore Exploration has only five exploration episodes as one First-Explore Exploitation follows each exploration. When sacrificial exploration is required (Fig. 3 B1-2), the controls catastrophically fail to explore, and learn to stay still instead B1: $RL^2$ (yellow) and VariBAD (grey) visit $< 2$ coordinates, and HyperX (teal), despite an exploration incentive, visits fewer coordinates than random action selection. First-Explore is unaffected and explores well and significantly better than random action selection (Fig. 3 B1 blue line above green line). Further, First-Explore exploration almost always visiting nine-unique coordinates in the first episode, and 8 more unique ones in the second. Provided this exploration context, First-Explore massively outperforms all other methods (Fig. 3 B2 orange line above all others). The kind-room results (Fig. C1-2) show that it is the fact that the Dark-Treasure Room requires *sacrificial exploration* that cripples the other methods, as all methods perform well when the average reward distribution is positive. However, First-Explore Exploitation still outperforms the others, highlighting the value of decoupling exploration and exploitation even in kind situations.

First-Explore's high training run consistency (e.g., tightly grouped coverage and reward in Fig. 3) suggests that the same systematic behaviour is being learnt regardless of network initialisation and training seed, suggesting First-Explore is learning something fundamental to the domain. This learning is promising as it potentially means First-Explore might learn a consistent algorithm or heuristic for general exploration if paired with a sufficiently-complex curriculum and task-distribution.

## 6 LIMITATIONS AND FUTURE WORK

As presented, First-Explore Exploration is myopic. The explore policy $\pi_{\text{explore},\theta}$ trains to most increase the expected reward of the explore-conditioned exploit policy $\pi_{\text{exploit},\theta}$ with each exploration. Unfortunately, iterated optimal myopic exploration does not necessarily produce an optimal sequence of explorations (Fig. 4). A potential solution is to reward exploration episodes with weighted sums of the rewards of all subsequent exploitation (analogous to summing discounted future reward in standard-RL).

Figure 4: Well-planned sequential exploration (left) and myopic exploration (right) of an area from the center over multiple episodes. The initial explore (red) is equally good, but myopic exploration hinders the second explore (green) as it must revisit previously seen locations, and more so for the third (purple).

Surprisingly, First-Explore is able to sometimes achieve higher *cumulative* reward than meta-RL algorithms that optimize for cumulative exploitation. Holding promise for using First-Explore to improve cumulative exploitation, e.g., initialize with a trained First-Explore policy, or a multi-headed architecture, etc...

While in the real world, true innovation often requires temporary sacrifice (and *sacrificial* exploration) it may be that RL has so far avoided such domains. There are however many RL applications that naturally decouple into no-stakes training/exploration, and high stakes performance/exploitation, and on these problems First-Explore should excel.

Another First-Explore limitation is that (seemingly no-stakes) rewards sometimes matter. E.g., if a chef robot is learning a new recipe in a physical home then it is vital the robot behaves safely and does not endanger humans or damage the kitchen while learning (unlike say, getting poor flavor or dish

presentation). First-Explore being willing to *sacrifice* reward could be dangerous, as it might ignore a safety reward penalty in order to master the recipe. One potential solution is to infinitely penalize endangerment or damage while training both the explore and exploit policy. This proposed version of First-Explore could actually result in far safer training (via in-context adaption) than attempting to use standard-RL, as the meta-RL policies would have learnt a strong prior of avoiding potentially endangering actions. However, it could be that such a strong penalty could prohibit effective training too. The issue is worth investigating. There are however many scenarios where exploration can be truly no stakes, e.g., any (sandboxed) simulation.

A final problem is the challenge of long sequence modelling, with certain environments requiring a very large context and high compute (e.g., can one have a large enough context, and enough compute to allow First-Explore to generalize and act as a replacement for standard-RL?). AdA (Adaptive Agent Team et al., 2023) hints such a project might be possible, and as progress on efficient long-context sequence modelling (Tay et al., 2020; Gu et al., 2021), research on RL transformer applications (Laskin et al., 2022; Chen et al., 2021), and Moore's Law all continue it becomes more feasible.

# 7 DISCUSSION

Given that First-Explore uses RL algorithms to train the meta-RL policy, how can it solve hard-exploration domains that standard-RL cannot? For example, how might First-Explore solve a sparse exploration problem (e.g. design a rocket for the first time). We believe that given a suitably advanced curriculum, and sufficient compute, First-Explore could learn powerful exploration heuristics (i.e., develop intrinsic motivations, e.g., an analogue of curiosity) and that these heuristics would enable such hard problems to be tackled (with great sample efficiency).

Curricula work by constantly challenging agents (or humans) with a suitable level of task. E.g., initially the First-Explore agent can only randomly explore and must learn to exploit based on random exploration. Once it has learnt rudimentary exploitation, the agent can learn rudimentary exploration. Then it can learn better exploitation, then better exploration, and so on, each time relying on there being 'goldilocks zone' tasks (Wang et al., 2019) that are not too hard and not too easy.

Further, while curricula can aid all of meta-RL, e.g., $RL^2$ and AdA, First-Explore can have a significant training advantage on certain problems (e.g., in the ten episode Dark Treasure-Room, First-Explore achieves positive cumulative reward while the standard meta-RL methods catastrophically fail). This advantage could potentially allow orders of magnitude greater compute efficiency, and allow otherwise infeasible curricula.

One might wonder how significant a limitation exploring by exploiting is, given that standard-RL seems to succeed despite it. We argue the difference is more significant, the more intelligent and human-like the adaptation. However, in both problem domains, the results show how optimal exploiting and exploring significantly differ, both in how they cover the state or action space, and in how and whether they help achieve high reward, and how this difference matters in order to achieve efficient in-context learning.

# 8 CONCLUSION

We identify the problem of attempting to explore by exploiting, and demonstrate that the novel meta-RL framework, First-Explore, solves this problem via the simple modification of learning two policies (one to first explore, another to then exploit). This paradigm of learned, intelligent exploration informing learned exploitation significantly improves meta-RL performance (even getting higher cumulative reward than state-of-the-art cumulative reward meta-RL). First-Explore performs better on even simple domains such as the multi-armed Gaussian-bandit, and massively improves performance on domains that require *sacrificial* exploration, such as the Dark Treasure Room environment.

We believe combining First-Explore with a curriculum, such as the AdA (Adaptive Agent Team et al., 2023) curriculum, could be a step towards creating algorithms able to exhibit human-level performance on unseen hard-exploration domains, which is one of the core challenges of creating artificial general intelligence (AGI). Provided we can adequately address the real and significant safety concerns associated with developing AGI, such developments would allow us to reap AGI's tremendous potential benefits.

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

## APPENDIX

## A    REPLICABILITY

For full transparency, replicability, and to make it easier for future scientists to build on our work, we have open sourced the training code, visualization code, the significance plots code, and environment code. The code is attached in the SI and anonymous for review. For the controls (VariBAD, $RL^2$, HyperX) we share environments, and the configuration files (that specify the training hyperparameters).

## B    COMPUTE

Each training run commanded a single GPU, specifically a Nvidia T4, and up to 8 cpu cores. Table 1 gives the approximate walltime of each run.

Table 1: Compute Usage Per Training Run

| Run | Runtime |
| --- | --- |
| Stochastic Bandit First-Explore | 18.5 hours |
| Stochastic Bandit $RL^2$ | 40 hours |
| Dark Treasure-Room First-Explore | 50 hours |
| Dark Treasure-Room HyperX & VariBAD & $RL^2$ | 10 hours |
| Dark Treasure-Room HyperX & VariBAD & $RL^2$ | 10 hours |

Notably, First-Explore was run for significantly longer on the Dark Treasure Room. First-Explore training was extended as performance continued to improve with additional training. Conversely, no change was observed in HyperX, VariBAD, or $RL^2$ (with VariBAD and $RL^2$ rapidly converging to a policy of staying still, while HyperX more slowly converged (as the exploration incentive was attenuated)). Similarly, the $RL^2$ Stochastic Bandit training was extended, as initial shorter runs were still improving, see Figure 5.

## B.1 Control Convergence

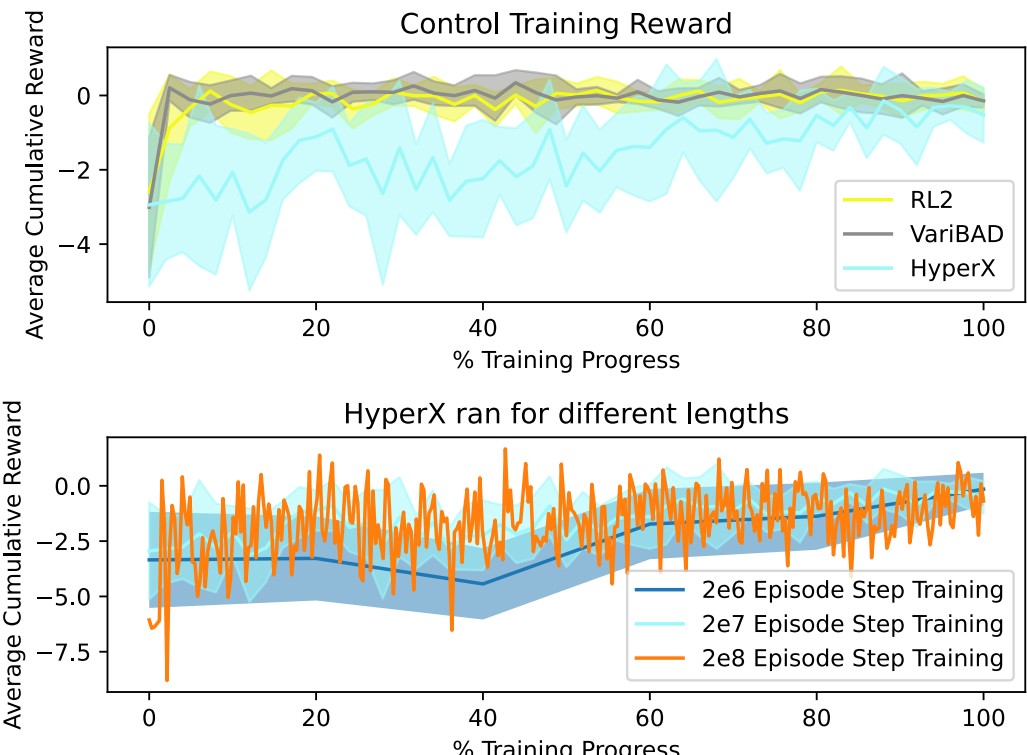

Figure 5: Control Training on the Dark Treasure-Room Environment. **Top:** $RL^2$, VariBAD, and HyperX average cumulative reward plotted against training time. $RL^2$ (yellow) and VariBAD (grey) converge to zero reward almost immediately. This transition corresponds to the policies learning to stay still (as the coverage plots demonstrate in Figure 3). HyperX (teal) reward increases (toward zero) throughout meta-training. However, this increase in reward comes not from HyperX learning an increasingly sophisticated policy, but instead is the result of the HyperX algorithm's meta-training exploration bonus being linearly reduced from the start to the end of meta-training. Thus, once that bonus is near zero, HyperX also learns to stay still. **Bottom:** HyperX with different training lengths (specified by number of episode steps). When HyperX is run for ten times as long (orange) or ten times less long (blue) than the default training time (light blue) the same behaviour is observed (of slow convergence to (slightly below) zero reward). This behaviour demonstrates the improvement in reward comes from the HyperX algorithm reducing the exploration incentive during the meta-training. It also implies that changing the length of training runs (including running for much longer) would not change the final performance results.

## C Evaluation Details

Evaluation (sampling the multiple evaluation environments and performing iterated rollouts) was with a single GPU.

For the Bandit Results each of the 100 First-Explore evaluations (10 for each seed) used a batchsize of $10,000$. For the 100 $RL^2$ Bandit evaluations batch size was reduced 2,000 (due to taking longer to evaluate), and since there is no training variance for UCB1 and TS ten evaluations were done, each with batchsize of $10,000$.

For the Dark Treasure Room and Kind Treasure Room results a evaluation of batchsize of $1,000$ was used for policies. All trained policies were evaluated 100 times (10 for each seed).

# D  TRAINING DETAILS

## D.1  CONTROLS:

VariBAD, HyperX and $RL^2$ were run via the official VariBAD (VariBAD and $RL^2$) and HyperX (HyperX) codebase. The hyperparameters they used for their gridworld environments were used, with the Dark Treasure-Room and Stochastic Bandit Environments being re-implemented to conform to the codebase (and tested to ensure the environments performed identically). See the SI attached configuration file for the exact hyper-parameters.

## D.2  FIRST-EXPLORE:

The architecture for both domains is a GPT-2 transformer architecture (Radford et al., 2019) specifically the Jax framework (Bradbury et al., 2018) implementation provided by Hugging Face (hug), with the code being modified so that token embeddings could be passed rather than token IDs. The different Hyperparameters for the two domains are given in Table 2.

For both domains each token embedding is the sum of a linear embedding of an action, a linear embedding of the observations that followed that action, a linear embedding of the reward that followed that action, a positional encoding of the current timestep, and a positional encoding of the episode number. See the provided code for details. For the dark treasure-room environments a reset token was added between episodes that contained the initial observations of the environment, and a unique action embedding corresponding to a non-action. The bandit domain had no such reset token.

Table 2: Model Hyperparameters

| Hyperparameter | Bandit | Dark |
|---|---|---|
| Hidden Size | 128 | 128 |
| Number of Heads | 4 | 4 |
| Number of Layers | 3 | 4 |

For training we use AdamW (Loshchilov and Hutter, 2019) with a piece-wise linear warm up schedule that interpolates linearly from an initial rate of 0 to the full learning rate in the first 10% training steps, and then interpolates linearly back to zero in the remaining 90% of training steps. Table 3 gives the optimization hyperparameters.

Table 3: Optimization Hyperparameters

| Hyperparameter | Value |
|---|---|
| Batch Size | 128 |
| Optimizer | Adam |
| Weight Decay | 1e-4 |
| Learning Rate | 3e-4 |

Hyperparameters were chosen based on a relatively modest amount of preliminary experimentation. Finally, for efficiency, all episode rollouts and training was done on GPU using the Jax framework (Bradbury et al., 2018).

## D.3  OPTIMIZATION LOSS

The First-Explore policies are trained by a novel optimization approach. To learn to exploit we learn the distribution of actions that lead to 'maximal' exploit episodes. Here we define an exploit episode as maximal if it a) has higher or equal reward to the best reward found in all of the previous First-Explore explore and exploit episodes in the current environment, and b) also exceeds a set baseline reward (hyperparameter) for the domain, see Algorithm 1. To learn to *explore* we learn the distribution of actions that lead to 'informative' explore episodes. Informative episodes are those that when added to the context lead to a subsequent exploit episode that a) exceeds the best

reward of previous First-Explore explore and exploit episodes and b) has higher reward than the environment baseline. This explore criterion is slightly different from the exploit 'maximal' criterion, as it requires an improvement in reward, see Algorithm 1. The baseline reward is there such that the first First-Explore exploit and explore episodes have an incentive to be respectively exploitative and exploratory.

Because in the dark treasure-room each episode is composed of multiple actions, the probability of an initial action leading to any outcome is potentially dependent on the distribution of future actions (e.g., imagine requiring two up actions to reach a reward; the first up action is no better than the first down action if the policy always moves down in the second step). Hence, one must learn the distribution conditional on a rollout policy. This expression is shown in Equation 1 for the case of the exploit distribution. Here "episode is maximal" refers to an exploit episode having higher reward than the baseline reward and the previous First-Explore explores and exploits (see previous paragraph). $a_t$ refers to the current action, and $[a]_{i>t} \sim \pi$ expresses how subsequent actions are taken under the rollout policy.

$$\mathbf{P}(\text{episode is maximal}|a_t, [a]_{i>t} \sim \pi) \tag{1}$$

To learn this distribution, the predicted likeihoods of actions being 'maximal' or 'informative' are compared to the action distributions of the rollouts that are 'maximal' or 'informative.' The predictions are improved by minimizing a cross entropy loss between the actions observed in the maximal and informative episodes, and the calculated probability of those actions being selected. This loss is detailed in Algorithm 1 as well as the provided code.

Once learned, the explore and exploit distributions combined with a sampling temperature each then specify a policy that with high probability selects actions likely to lead to good exploitation or good exploration. To ensure that all actions are sampled and to provide more exploration during training (of both the explore and exploit policy), we add a small probability $\epsilon$ chance of selecting a random action instead of one sampled from the unmodified explore or exploit policy. This probability is then a hyperparameter that can be tuned. Learning the distributions then allows iteratively updating the rollout policies by each time taking the new rollout policies and learning the new distributions of maximal and informative actions under the rollout policy. The frequency of such updates is then a hyperparameter. The hyperparameters used are given in Table 4.

While preliminary experiments found this meta-RL training method performed best, we believe the First Explore meta-RL framework will work for general approaches too, such as using policy gradient with actor critic, or Muesli (Hessel et al., 2021) which was used in AdA (Adaptive Agent Team et al., 2023).

Table 4: Training Rollout Hyperparameters

| Hyperparameter | Bandit | Darkroom |
|---|---|---|
| Exploit Sampling Temperature | 1 | 1 |
| Explore Sampling Temperature | 1 | 1 |
| Policy Update Frequency | every training step | every $10,000$ training steps |
| $\epsilon$ chance of random action selection | 0.05 | 0 |
| Baseline Reward | 0 | 0 |
| Training Updates | 200,000 | 1,000,000 |

For evaluation, we then sample by taking the argmax over actions, and do not add the $\epsilon$-noise.

```
# rollout conducts an episode when provided with an environment and policy
# and returns all the episode infomation
def model_conditional_actions(θ, π, baseline_reward):
    # sample an environment, and initalize context c and loss values
    m = sample(𝓜); c = set(); loss = 0
    best_reward_seen = baseline_reward
    for i in range(k): # do k iterated rollouts
        τ_explore = rollout(m, π_explore,c)  # explore given context c
        τ_exploit = rollout(m, π_exploit,c ∪ {τ_explore})  # exploit given c ∪ {τ_explore,}
        r = final_reward(τ_exploit) # get the exploit reward
        # calculate a weight on the episodes
        # non-increasing episodes have zero weight
        # and increasing episodes have weight proportional to reward improvment
        explore_weight = 1_{r>best_reward_seen} * (1 + r − best_reward_seen)
        exploit_weight = 1_{r≥best_reward_seen} * (1 + r − best_reward_seen)
        explore_loss = cross_ent(π and θ predicted probability, τ_explore actions)
        exploit_loss = cross_ent(π and θ predicted probability, τ_exploit actions)
        # update the loss, conditional on the episodes being improvements
        loss = loss + explore_weight * explore_loss
        loss = loss + exploit_weight * exploit_loss
        c = c ∪ {τ_explore}  # update the context for the next explore
        # update the best reward seen
        best_reward_seen = max(best_reward_seen, final_reward(τ_exploit), final_reward(τ_explore))
    return loss
```

Algorithm 1: Training to model conditionally increasing exploits with First-Explore rollouts.

## E  DARK TREASURE-ROOM VISUALIZATIONS

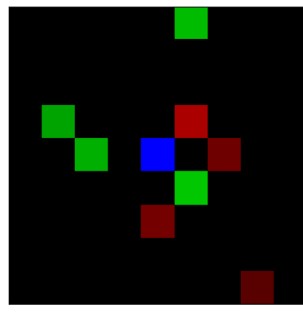

Figure 6: A visualization of the dark treasure-room. The agent's position is visualized by the blue square, positive rewards are in green, and negative rewards are in red, with the magnitude of reward being visualized by the colour intensity. When the agent enters a reward location it consumes the reward, and for that timestep is visualized as having the additive mixture of the two colours.

Here are example iterated First-Explore rollouts of the two trained policies, $\pi_{explore}$, $\pi_{exploit}$, visualized for a single sampled darkroom.

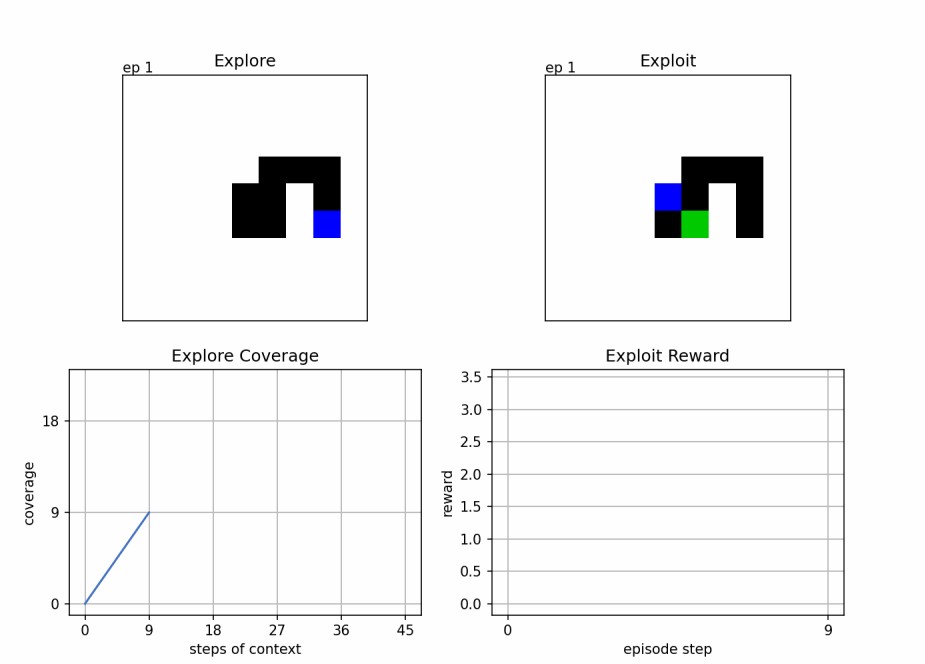

Figure 7: The first (First-Explore) explore episode. **Top left** visualizes the last step of a First-Explore explore episode, with the locations that are not in the cumulative context being coloured white, as the agent is blind to them (having no observations or memory of those locations). This figure plots the end of the first exploration, and shows a reward has been found. **Bottom left** visualizes the coverage of the cumulative context by plotting the total number of unique locations visited by the exploration against the cumulative episode step count. In this explore, the agent never doubled back on itself, which is good as it is optimal to have as many unique locations visited as possible. **Top right** visualizes a step in a First-Explore exploit episode, with the locations that are in context visualized. The agent can effectively 'see' those locations in its memory. **Bottom right** plots the exploit reward against the exploit episode timestep. As this figure plots before the start of the exploit episode, the agent has yet to move and encounter rewards, but will have done so in the subsequent visualizations.

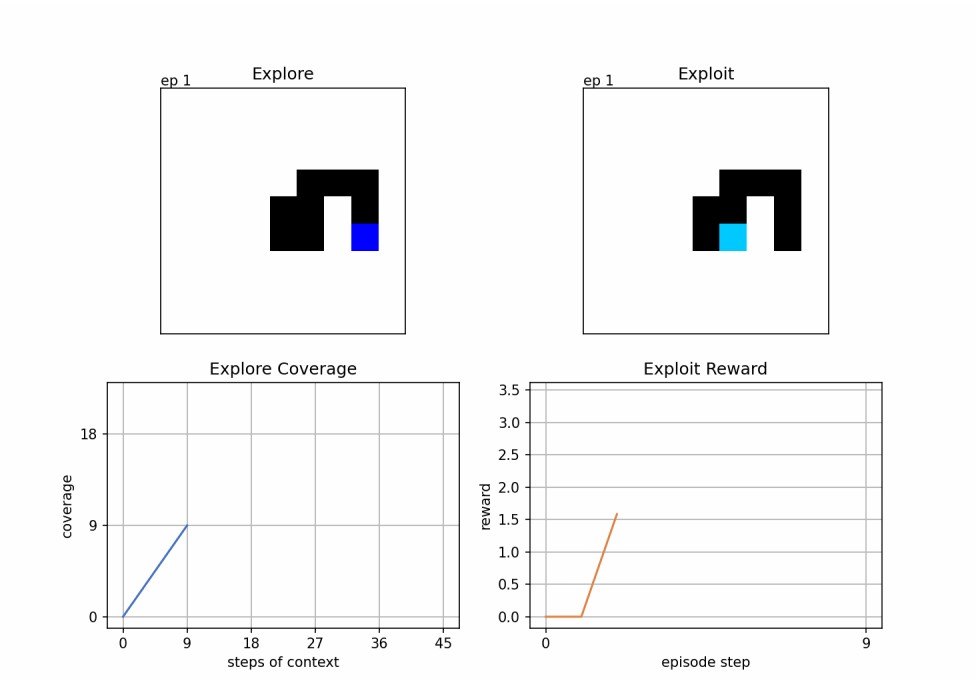

Figure 8: The first (First-Explore) exploit episode. This figure uses the same visualization design as Figure 7. **Left top and bottom** are the same as in Figure 7, and of the explore context, not the current exploit episode. **Right top**, the agent (the light blue square) has found the reward in the first two steps. Consuming the reward is visualized by the agent color and the reward color being combined. **Right bottom**, the associated episode reward is shown.

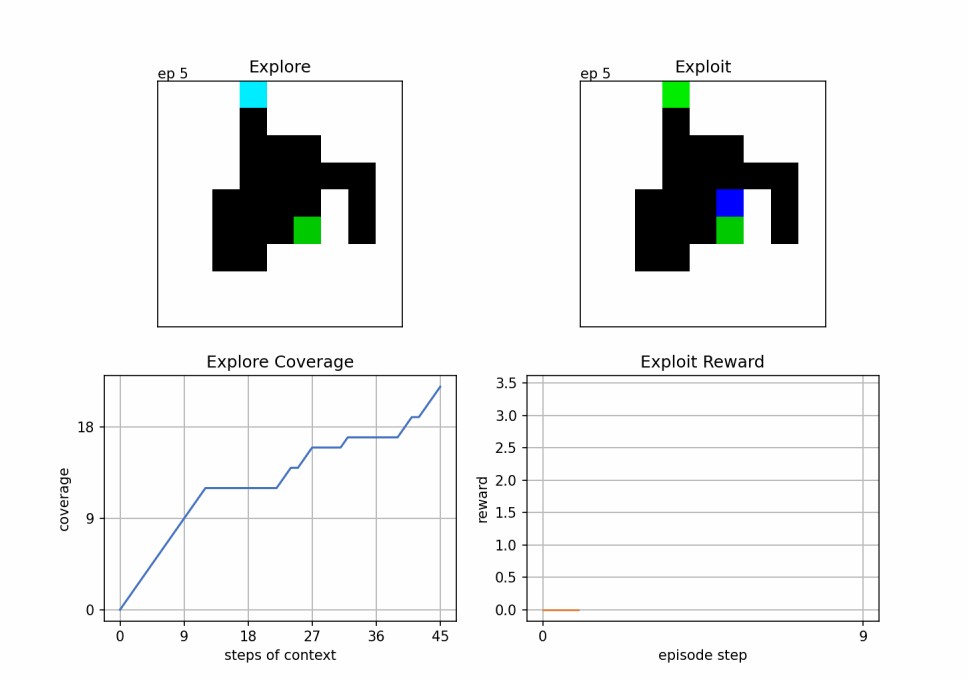

Figure 9: The fifth (First-Explore) explore episode. At the end of the 5th explore episode the agent has discovered a new positive reward at the top of the room, and can now 'see' it in memory. The new information presents an opportunity for the exploit policy to obtain both rewards, but it only has exactly enough time-steps in an episode to navigate to do so, and thus cannot make a mistake navigating.

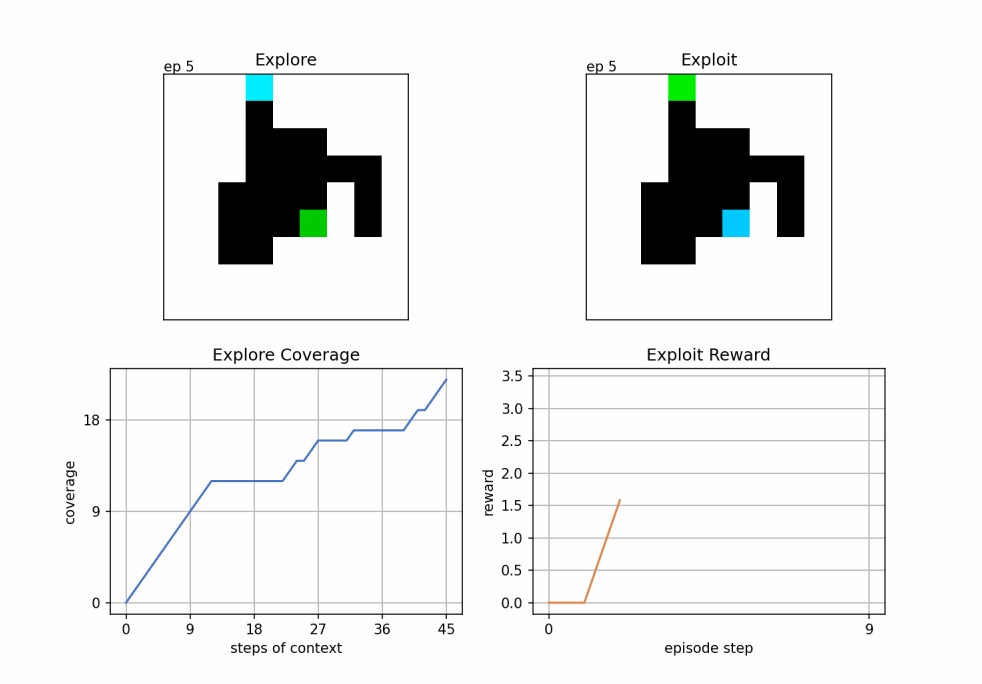

Figure 10: The first reward of the fifth (First-Explore) exploit episode. Two steps into the episode the agent (in consuming, light blue) has consumed the nearby reward.

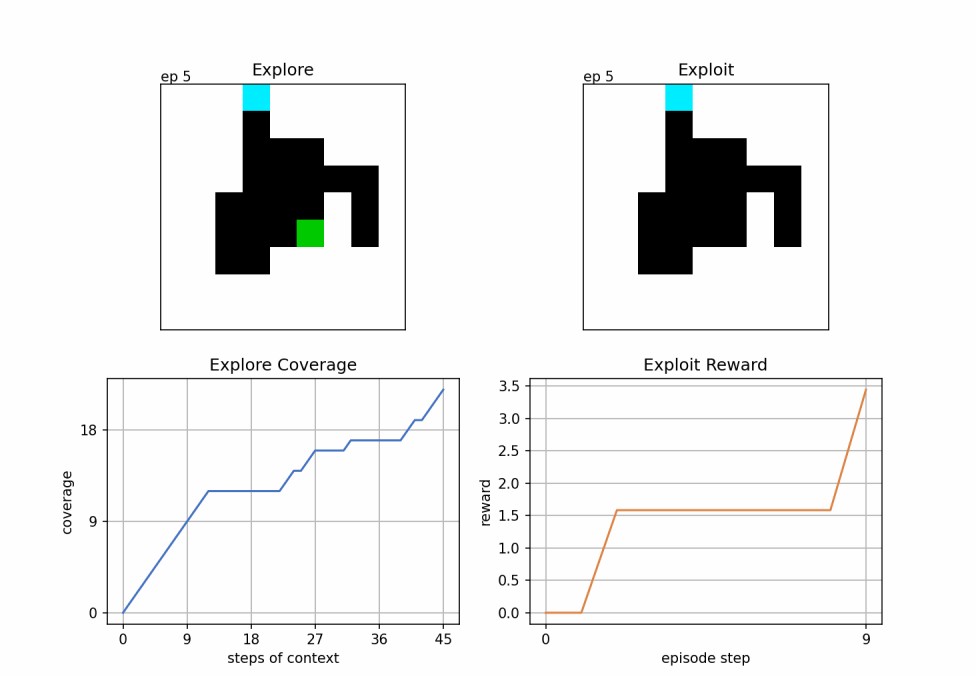

Figure 11: The end of the fifth (First-Explore) exploit episode. After consuming the nearby reward the agent has reached the newly discovered reward at the top of the room and consumed it. This success required making no mistakes and pathing first to the nearby reward then to the top one on the first try. This inference is possible because the quickest the agent can reach both rewards is exactly the length of the episode (9 steps). The pathing in this episode is an example of intelligent exploitation, as after the information reveal (the reward at the top) of a single episode the agent appropriately changes its policy based on the context and using the learnt environment prior (e.g., how to navigate), produces a highly structured behaviour (pathing with no mistakes).

