# OpenReview forum: "First-Explore, then Exploit: Meta-Learning Intelligent Exploration"
_ICLR.cc/2024/Conference — ICLR 2024 Conference Withdrawn Submission_

### Official Review · Reviewer_tz8Y · 2023-10-30

**Soundness:** 3 good
**Presentation:** 3 good
**Contribution:** 3 good
**Rating:** 8
**Confidence:** 3

**Summary:**

The paper proposes a new meta-RL scheme, which decouples exploration and exploitation separately, and optimizes exploration to maximize the reward of the subsequent exploit while optimizing exploitation to maximize episode reward. By doing so, the exploration policy can perform purely sacrificially exploration but still be grounded on the exploitation reward. The proposed method outperforms baselines in bandit and 9x9 treasure room domain. Authors also provided impartial limitations and discussions.

**Strengths:**

1. The idea is intuitively well motivated, and clearly demonstrated. Paper is well-written and easy to follow.

2. Code of the proposed method, configurations of baselines and environments are all available.

**Weaknesses:**

1. Although the idea is intuitively well motivated, it might still be good to have some experimental/theoretical motivations behind the intuition.

2. Minor comments:

a. In Fig.3, there is too much space in the fifth row in the caption.

b. Structure-wise, I feel like section 3 can be splitted into 2 different subsections, so that readers can have a clearer overview of it. For example, two subsections could be “meta-rl” subsection and “other works addressing exploration”. Of course you don’t have to change.

**Questions:**

You discussed DREAM in the related work, which also decouples exploration and exploitation (only explores for one episode then starts exploitation). Why didn’t you compare it with DREAM as well? Do you think in tasks you performed, only exploring for one episode is enough?

---

### Official Review · Reviewer_fuX7 · 2023-10-31

**Soundness:** 2 fair
**Presentation:** 1 poor
**Contribution:** 2 fair
**Rating:** 3
**Confidence:** 3

**Summary:**

This paper addresses the exploration-exploitation trade-off in meta-RL. Specifically, it proposes the First-Explore framework, which separates the exploration and exploitation stages to achieve faster adaptation. Iteratively, the exploration policy is first rolled out, whose trajectories are used as the context of the exploitation policy. Then, the exploitation policy is rolled out, and the reward of the resulting trajectory will be used as the signal to train the exploration policy. With this mechanism, the exploration policy is incentivized to collect information for achieving high rewards while not being punished by potential negative rewards. Empirical results show that First-Explore can learn effective exploration policies in simple multi-arm bandit and treasure-finding domains, outperforming other meta-RL algorithms.

**Strengths:**

The strength of the work is its originality and significance. To the best of my knowledge, the idea of the paper is novel, and it seems to be the only meta RL algorithm that can explore efficiently in tasks with punitive rewards for exploration.

**Weaknesses:**

However, the writing of the paper is not ready for publication, which hinders readers from understanding it easily and is its biggest weakness:
* First of all, the paper should be reorganized and have more details of the proposed framework in the main text. The current version only has a vague description of the framework in the main text, which doesn’t provide adequate information for the reader to understand the method. I suggest the authors should at least describe a complete instance of the proposed framework in the main text.
* Second, the paper is full of typos and grammatical/format errors, which create significant barriers for the readers. See below for a few examples before pape 5:
1. In the second paragraph of the Introduction, there should be a comma before “e.g.”
2. In the definition of coincidental exploration, “exploitation” should be “exploration”
3. On page 2, “slow, and” should be “slow and”
4. On page 2, “In the dungeon” should be followed by a comma. Same for “while exploring” and “with the fixed dungeon” on the same page
5. On page 2, “be will” should be “will be”
6. On page 3, “Each training step” should be “In each training step”
7. On page 3, “has memory of” should be “has the memory of”
8. On page 3, “the exploratory episodes number” should be “the exploratory episode number”
9. On page 4, “after every explore providing” should be “after every exploration,providing”
10. On page 4, there are $\pi_{\text{explore},\theta,c}$, $\pi_{\text{explore},\theta|c}$. Be consistent with the notation.
11. On page 4, how is the expected reward of an episode defined? Also, it seems that the reward part is missing in $\mathbb{E}(\tau)$.
12. On page 5, “While an innovation” is not a complete sentence.
13. On page 5, “as replacement” should be “as a replacement”
14. On page 5, $k$ is not defined in “\mu_{\{1,\dots,k\}}”
15. On page 5, “(Fig. 2 A b)” should be “(Fig. 2 A), b)”
16. On page 5, “(c)” should be “c)”

Other minor suggestions
1. Consider adding a pseudocode of an instance of the proposed algorithm in the main text.
2. Consider breaking down some long sentences to make them easier to understand.

In addition, I have some questions about the experiments. See the Questions below.

**Questions:**

1. Does the x-axis in the plots of Figure 2 include the arm pulls from both the exploration and exploitation policies? It seems that it doesn’t. If this is the case, how is the comparison to UCB1 (and maybe other algorithms) fair?
2. During training, how does First-Explore handle the context?
3. How sensitive is the proposed method to the episode length? It may be beneficial to investigate this to understand the applicability of the method.

---

### Official Review · Reviewer_obj6 · 2023-11-01

**Soundness:** 2 fair
**Presentation:** 1 poor
**Contribution:** 2 fair
**Rating:** 3
**Confidence:** 3

**Summary:**

In First-Explore, then Exploit: Meta-Learning Intelligent Exploration, the authors introduce a novel method of meta-learning focused on the idea of alternating explorative and exploitative policies to enable what they call “sacrificial exploration”. They argue that a major problem with the current literature in Reinforcement Learning is that they attempt to explore and exploit simultaneously, and that their method, which employs a policy whose focus is exploration, eliminates this issue.

**Strengths:**

The paper proposes a cogent analysis of exploration in the modern Reinforcement Learning literature, and certain of its weaknesses, through a well-written review of the recent literature. It  introduces a reasonable framework for a meta-learning algorithm and demonstrates that using that framework, significant improvement in average rewards and state coverage can be achieved.

**Weaknesses:**

The paper’s description of the novel method is extremely lacking in detail, to the point that it is not possible to even sketch pseudocode for their idea from the relevant section. Their analysis of exploration in the present literature, while cogent, lacks serious consideration of the benefits of what they call “coincidental exploration”, especially as pertains to traditional frameworks for exploration centered on regret minimization. The meta-learning aspect of the paper is also somewhat doubtful—given the modest description present in the paper, it appears that the “meta”-RL policy which is learned is in fact simply a pair of transformer-based models, each of which act based upon the context of the trajectories seen in a given “training run”. That is not, to my knowledge, the usual sense of the term “meta-RL”. The paper also fails to conduct fair experiments, running the proposed algorithm for 5x as long as the comparators, without adequate justification. Further, this fact is obscured into the appendix. In my view, this behavior represents an attempt to fraudulently represent the proposed algorithm.

**Questions:**

1. When the paper claims that First-explore is capable of achieving higher cumulative reward, is this across the training of the meta-RL policy, or only on the last run of the RL method?
2. What is the view of the authors on exploration methods which work based upon exploration bonuses? Are such methods sacrificial, or coincidental? These methods certainly seem capable of sacrificing at least some degree of reward for the purpose of exploration. Are these not “standard RL”?
3. What is the basis of the claim that “Only meta-RL is both a) computationally tractable and b) can potentially achieve human-level sample efficiency in complex environments.”?
4. Did the authors try any tasks where, concordant with the claim that “On complex tasks, the policies need to be learnt together”, the policies did not need to be learned together?
5. Why was the fact that First-Explore used a transformer reserved for the results section?
6. How does the “novel loss based on predicting the sequence of future actions conditional on the episode having high reward, which preliminary experiments showed improved training stability.” relate to the loss function of Upside Down Reinforcement Learning (Shmidhuber, 2020) (https://arxiv.org/abs/1912.02875)?
7. Does “hand-coded strategy of picking the arm with the highest mean reward seen” mean the same thing as “greedy policy with respect to observed rewards”?
8. Can the authors explain why myopically optimal exploration would generally be worse than the so-called “optimal sequence of explorations”? It would seem that there could be many reasons to prefer the myopic strategy.
9. I am concerned that this work is more akin to previous works on multi-task RL, wherein a policy is trained to be able to perform many tasks, than to meta-RL, where (to my knowledge) it is typical to train the new policy in a very significant way, not just by changing the context of a transformer model. Would the authors address this concern?

---

> ### Author Response · Authors · 2023-11-22
>
> Thank you for your response. As described in the general response, while we think a revised version of the paper makes a valuable contribution, we think it is unlikely we would be able to convince reviewers to change their scores enough to lead to publication given the initial set of scores. We are thus withdrawing the paper to improve it based on the feedback generously provided by the reviewers. That said, we do want to respond to one of your statements for the record.
>
> We disagree in the strongest terms with your claim that the paper “represents an attempt to fraudulently represent the proposed algorithm.” We think it is improper for you to have made this serious accusation given that, rather than hide the training time disparity between First-Explore and the controls, we explicitly mentioned it in black and white! It is true that it is in the appendix, but the appendix exists for the public record and is an important and effective form of scientific disclosure (as evidenced in part by the fact that you learned this information in the paper’s appendix). It is true that it being in the appendix indicated we thought it is a less important detail, and that was because there are straightforward technical reasons and empirical results that allow us to be confident that running the controls (which achieve at best zero performance on average) would not perform above zero with greater training, and that they thus far underperform First-Explore. We next quote the text from the appendix that clearly described what we did and why. We then describe in even more detail the reasons we can be confident the results of the controls would not change if run longer (including new plots, which also support this conclusion). That said, we agree it would be better to summarize these results in the main paper and include plots of them in the appendix (see below for more discussion on these points), which we will do in future versions of the paper.
>
> Quoting from the appendix, “Notably, First-Explore was run for significantly longer on the Dark Treasure Room. First-Explore training was extended as performance continued to improve with additional training. Conversely, no change was observed in HyperX, VariBAD, or RL2 (with VariBAD and RL2 rapidly converging to a policy of staying still, while HyperX more slowly converged (as the exploration incentive was attenuated)).” Compute costs money, time, carbon, and other valuable resources; if it is clear that an algorithm will not improve performance with more training, we feel it is acceptable to decide not to waste compute watching it stay at zero if, critically, one describes that fact and explains the choice clearly, which we did.
>
> We have added plots detailing this training behavior. The plots are shown in the revised paper. In the plots (Figure 5 Top subplot in the appendix) one can see that VariBAD and RL2 quickly converge to receiving zero reward. These training results agree with the coverage results described in the main paper that show how VariBAD and RL2 have learnt to stay still (and once having learnt to stay still it is prohibitively hard to learn good cross episode exploration and exploitation).
>
> The HyperX results require more information to properly interpret because HyperX reward increases (toward zero) throughout meta-training. However, this increase in reward comes not from HyperX learning an increasingly sophisticated policy, but instead is the result of the HyperX algorithm meta-training exploration bonus being linearly reduced from the start to the end of meta-training. At the beginning of training there is a large bonus for exploration, and at the end zero bonus. Hence, towards the end of meta-training HyperX converges towards staying still (and thus a reward of zero, like VariBad and RL2). Figure 5 bottom subplot of the appendix shows a comparison between different HyperX meta-training runs that demonstrates that changing the number of training steps (a hyper-parameter of the training due to the linear scaling of the exploitation bonus) does not qualitatively change the result (i.e. reward anneals from negative to zero, arriving around zero by the end of training, no matter how long the training run is). Based on this understanding of the dynamics and the results observed, we are confident that running for even longer would produce the same dynamic. However, we do agree it would be better for these facts to be summarised clearly in the main text and have plots of the controls in the appendix. That we did not do so in the submitted version occurred due to the rush to meet the submission deadline, and we will fix these issues in future versions of the paper.

---

### Official Review · Reviewer_RsHA · 2023-11-06

**Soundness:** 2 fair
**Presentation:** 2 fair
**Contribution:** 2 fair
**Rating:** 3
**Confidence:** 4

**Summary:**

The paper proposes an approach for meta-exploration by having separate explore and exploit policies. The context collected by the explore policy is provided to the exploit policy, and the explore policy is optimized to maximize the exploit policy reward.

**Strengths:**

While meta-exploration for RL is an important problem, there are issues/concerns I have with the current version of the paper, please see weaknesses.

**Weaknesses:**

1. Comparisons and environments

The paper claims that the presented method can effectively meta-explore, but I have numerous questions regarding the experiments. How does the full proposed approach perform on standard meta-RL benchmarks [1], and how does it compare to meta-rl algorithms like pearl [1] and RL2 [2] on these environments? Is the sample efficiency better because of enhanced exploration, or is there a tradeoff? The paper proposes separate exploration and exploitation policies for meta-learning, how does this compare to meta-cure which also uses a similar approach [3], on the cheetah/walker/hopper meta-envs from that paper ?

2. Idea formulation

The idea of learning how to collect data/explore in order to get sufficient context to subsequently exploit is very common in previous meta-RL algorithms ([1,2,3]). The approach of having explicitly separate policies for exploration and exploitation has also been studied, showing distinct information-seeking behaviors in the exploration phase [3]. Given this, the main difference of the proposed approach is the use of the transformer architecture, but this choice isn't ablated, so it is unclear if this is really important/effective for RL control tasks.


[1] : Rakelly, Kate, et al. "Efficient off-policy meta-reinforcement learning via probabilistic context variables."
[2] : Duan, Yan, et al. "Rl $^ 2$: Fast reinforcement learning via slow reinforcement learning."
[3] : Zhang, Jin, et al. "Metacure: Meta reinforcement learning with empowerment-driven exploration."

**Questions:**

Please address the questions in the weaknesses section

---

### Author Response · Authors · 2023-11-22

Thank you greatly for reviewing. We appreciate the time you have spent. We have closely read the reviews and considered all questions asked. We feel the paper makes an important contribution and there are good answers to your many great questions. However, we feel it is unlikely the scores would be sufficiently changed to lead to publication. We thus feel that the best path is for us to withdraw the paper to create more time for improving it based on your feedback, including making it clearer and adding new experiments. We hope you will consider reading the revised version of the paper when it is published in the future. Thank you again.